# Is dual use of nicotine products and cigarettes associated with smoking reduction and cessation behaviours? A prospective study in England

Sarah E Jackson ![ORCID] , Emma Farrow, Jamie Brown, Lion Shahab

Department of Behavioural Science and Health, University College London, London, United Kingdom

**Correspondence to**
Dr Sarah E Jackson;
s.e.jackson@ucl.ac.uk

## ABSTRACT

**Objectives** To investigate associations of dual use of e-cigarettes and cigarettes with subsequent quitting activity (smoking reduction, quit attempts and use of evidence-based cessation aids). To overcome potential confounding by factors associated with use of pharmacological support, we selected dual use of over-the-counter nicotine replacement therapy (OTC NRT) and cigarettes as a behavioural control.

**Design** Prospective cohort study with 6-month follow-up.

**Setting** England, 2014–2016.

**Participants** 413 current smokers participating in the Smoking Toolkit Study, a representative survey of adults in England, who reported current use of e-cigarettes or OTC NRT and provided data at 6-month follow-up.

**Main outcome measures** The exposure was dual use of e-cigarettes or OTC NRT at baseline. Outcomes were change in cigarette consumption, quit attempts and use of evidence-based cessation aids during quit attempts over 6-month follow-up. Relevant sociodemographic and smoking characteristics were included as covariates.

**Results** After adjustment for covariates, dual e-cigarette users smoked two fewer cigarettes per day at follow-up than at baseline compared with dual OTC NRT users (B=2.01, 95% CI −3.62; −0.39, p=0.015). While dual e-cigarette users had 18% lower odds than dual OTC NRT users to make a quit attempt at follow-up (risk ratio (RR) 0.82, 95% CI 0.67 to 1.00, p=0.049), the groups did not differ in use of cessation aids (RR 1.06, 95% CI 0.93 to 1.21, p=0.388).

**Conclusions** Dual use of e-cigarettes is associated with a greater reduction in cigarette consumption than dual use of OTC NRT. It may discourage a small proportion of users from making a quit attempt compared with dual OTC NRT use but it does not appear to undermine use of evidence-based cessation aids.

## Strengths and limitations of this study

► The use of a prospective study design allowed us to examine changes over time.
► However, the follow-up rate was low, and significant differences were noted between responders and non-responders on several variables.
► The reliance on self-reported data may increase measurement error or bias in key variables, such as quit attempts at 6-month follow-up.
► While analyses were adjusted for several potential confounders, there is the possibility of residual confounding by unmeasured variables.

aid and as a method of harm reduction among smokers not trying to quit.[3 4] As such, dual use of e-cigarettes and cigarettes ('dual e-cigarette use') is prevalent, with around 20% of the English smoker population also using e-cigarettes.[5] A growing body of evidence shows that e-cigarettes are substantially less harmful than cigarettes[6–9] and effective in promoting cessation.[10–14] However, concerns have been raised regarding unintended population-level consequences, including perpetuating nicotine addiction and reducing motivation to quit smoking,[15 16] which may undermine the positive impact of e-cigarettes on population health.[17]

Large-scale observational studies have examined associations between dual e-cigarette use and quit attempts in a real-world setting. These have largely,[10 18 19] but not exclusively,[11] observed a positive association, with e-cigarette use associated with a higher rate of quit attempts compared with non-use. There is also some evidence that e-cigarettes may help smokers to reduce the amount that they smoke. Cigarette consumption is a useful proxy for nicotine addiction, given nicotine addiction drives cigarette consumption[20] and smokers who are more addicted typically smoke a greater number of cigarettes

## INTRODUCTION

Cigarette smoking is a leading cause of morbidity and premature mortality, responsible for 96 000 deaths each year in the UK.[1] While substantial progress has been made in reducing smoking prevalence over recent years, around 15% of the UK adult population continue to smoke.[2] E-cigarettes are popular among smokers both as a smoking cessation

per day.[21] In a Cochrane review of available evidence on e-cigarettes and smoking reduction, all included prospective cohort studies documented substantial reductions in cigarette consumption among e-cigarette users relative to non-users.[22] Experimental studies have shown similar effects even in smokers not intending to quit.[23 24] This is notable given that many dual e-cigarette users do not use e-cigarettes for cessation.[25 26]

These findings suggest that a potential negative impact on quitting activity from the dual use of e-cigarettes and quitting activity appears unlikely. However, limitations of the existing literature include retrospective assessment of e-cigarette use[18] and lack of adjustment for key confounders, such as motivation to quit.[27 28] Moreover, the comparison groups of most previous studies consisted of smokers not using e-cigarettes. This is not a suitable real-world control due to risk of confounding, for example, by the comparatively higher levels of nicotine dependence and motivation to quit among smokers who use cessation aids.[29]

Alternative licensed non-combustible nicotine products, such as over-the-counter nicotine replacement therapy (OTC NRT), provide a more appropriate comparator. Until the recent surge in popularity of e-cigarettes, NRT was the most commonly used quitting aid in England.[30] Like e-cigarettes, NRT is widely available to purchase without prescription and provides nicotine to help reduce cravings and withdrawal symptoms in periods of abstinence. Previous real-world comparisons of e-cigarette and OTC NRT have focused on use for, and outcomes of, smoking cessation,[31] and supported the effectiveness of e-cigarettes. Further, a randomised controlled trial of e-cigarettes compared with OTC NRT and placebo e-cigarettes in smokers motivated to quit found that e-cigarette users were more likely to achieve reduced cigarette consumption and biochemically verified cessation than OTC NRT or placebo e-cigarette users.[32] Outside of a quit attempt, dual use of OTC NRT and cigarettes ('dual OTC NRT use') has been found to increase the real-world likelihood of quit attempts but not smoking reduction.[33] Understanding how dual e-cigarette use compares with dual OTC NRT use in its effects on smoking reduction and quit attempts in the real world, outside of clinical trials, is important for contextualising concerns about the risks of e-cigarette use.

Additionally, whether dual e-cigarette use is associated with cessation aid use in future quit attempts has significant implications for e-cigarettes' long-term impact. The use of cessation aids appears to promote quit success.[14 34] Thus, if the likelihood of cessation aid use in future quit attempts is increased, dual use of e-cigarettes would impact positively on population cessation rates. Currently, little is known of the association between use of aids on one occasion and their future use, although studies have found that use of medication (varenicline, bupropion, NRT) during a previous quit attempt is associated with a preference for the use of medication during a future quit attempt[35] and the use of the same cessation aid across

quit attempts.[36] As such, use of e-cigarettes might be associated with greater likelihood of using cessation aids to support a future quit attempt.

The present study therefore aimed to examine prospective associations of dual e-cigarette use versus dual OTC NRT use, with subsequent smoking reduction, quit attempts and use of evidence-based cessation aids. Specifically, we addressed the following research questions:
1. Is dual e-cigarette use at baseline associated with (a) smoking reduction and (b) quit attempts at 6-month follow-up compared with dual OTC NRT use?
2. Among those attempting to quit at the 6-month follow-up, is dual e-cigarette use at baseline associated with the use of evidence-based cessation aids compared with dual OTC NRT use?

## MATERIALS AND METHODS
### Design
A prospective cohort study with data obtained from the Smoking Toolkit Study was conducted. The Smoking Toolkit Study is an ongoing national study of smoking prevalence and patterns in England. The Smoking Toolkit Study protocol has been described fully elsewhere,[37] but briefly, it involves monthly face-to-face computer-assisted interviews with representative samples of adults in England. Households are selected using a hybrid of random location and simple quota sampling. From an initial approximately 170 000 grouped output areas, each comprising of approximately 300 households and stratified by Acorn characteristics (http://www.caci.co.uk/acorn/), grouped output areas are randomly selected and trained interviewers conduct interviews until quotas are reached. Approximately 1700 adults (≥16 years) are interviewed at baseline per month (one per selected household), with samples shown to be representative of the population in England with regard to smoking prevalence and sociodemographic characteristics.[37] In certain waves (due to availability of funding), respondents consenting to re-contact at baseline have been followed up by telephone survey 6 months later. All participants provided fully informed consent.

### Sample
For the present study, we analysed data collected during baseline survey waves 90 (March 2014) through 120 (September 2016), which included a 6-month follow-up survey. We restricted our analytic sample to those participants who (1) reported smoking cigarettes (manufactured or hand-rolled) daily or occasionally at baseline, (2) reported using either e-cigarettes or OTC NRT at baseline and (3) completed the 6-month follow-up survey.

### Public and patient involvement
No patients or members of the public were involved in setting the research questions or the outcome measures, nor were they involved in the design and implementation of this specific study. There are no plans to involve patients in dissemination.

## Measures

### Explanatory variable

The explanatory variable was dual use of e-cigarette or OTC NRT (nicotine gum, nicotine patch, nicotine inhaler, nicotine lozenges/tablets). This was measured at baseline with three questions:

1. Which, if any, of the following are you currently using to help you cut down the amount you smoke?
2. Do you regularly use any of these in situations when you are not allowed to smoke?
3. Can I check, are you using any of the following at all for any reason?

The response options for each of these questions were: nicotine gum; nicotine lozenge; nicotine patch; nicotine inhaler/inhalator; another nicotine product; electronic cigarette; nicotine mouthspray and other.

A binary variable was created, combining responses to these questions to distinguish between smokers who reported current use of e-cigarette and those who reported current use of NRT. Those using neither (73.5% of available sample) or both products (2.8%) were excluded.

### Outcome variables

Our outcomes of interest were: (1) smoking reduction (change in number *of* cigarettes <u>smoked</u> per day) between baseline and 6-month follow-up, (2) quit attempts between baseline and 6-month follow-up, (3) use of cessation aids during the most recent quit attempt among those who tried to quit between baseline and 6-month follow-up.

Cigarette consumption was assessed by self-reports of the number of cigarettes the participant usually smoked per day or per week, according to their preference. Cigarettes per day were calculated as the number of cigarettes smoked per day or the number of cigarettes smoked per week divided by seven. Change in cigarettes per day was analysed as a continuous variable computed as 6-month follow-up cigarettes per day minus baseline cigarettes per day. Participants who reported being a non-smoker at follow-up were excluded from analyses of smoking reduction (n=77).

Quit attempts between baseline and 6-month follow-up were assessed with the question: *How many serious attempts to stop smoking have you made in the last 6 months? By serious I mean you decided that you would try to make sure you never smoked again.* Quit attempts were coded 1 for those who reported at least one quit attempt and 0 for those who reported no quit attempts between baseline and follow-up.

Among those who made a quit attempt, use of evidence-based cessation aids (e-cigarettes, OTC NRT, face-to-face behavioural support and prescribed medication [NRT, bupropion or varenicline]) during the most recent attempt was assessed with the question: *Which, if any, did you try to help you stop smoking during the most recent serious quit attempt?* Response categories were collapsed to a binary variable indicating whether the quit attempt was aided or unaided.

### Covariates

Potential confounding variables adjusted for in the analyses were selected a priori and measured at baseline.

Sociodemographic variables included: age, sex and social grade. Age was analysed as a categorical variable banded by SD, to account for the non-linear association between smoking and age. Social grade is an occupational index of socioeconomic position closely linked with smoking behaviour,[38] categorised as ABC1, which includes managerial, administrative and professional and occupations, versus C2DE, which includes semi-routine and routine occupations, manual occupations, never workers and long-term unemployed.[39]

An index of nicotine dependence measured at baseline was included as an additional adjustment in the cigarettes per day change analysis due to the positive correlation between levels of dependence and cigarettes per day.[21] This was measured with the item: *How much of the time have you felt the urge to smoke in the past 24 hours?* The response options ranged on a continuous scale from 0 (*not at all*) to 5 (*extremely strong*).

Motivation to quit smoking measured at baseline was included as additional adjustment in the quit attempt analysis as it has been shown to be highly predictive of quit attempts.[27 28] This was measured using the Motivation to Stop Scale,[40] a single item with 7 response categories coded as: (1) *I don't want to stop smoking*; (2) *I think I should stop smoking but don't really want to*; (3) *I want to stop smoking but haven't thought about when*; (4) *I REALLY want to stop smoking but I don't know when I will*; (5) *I want to stop smoking and hope to soon*; (6) *I REALLY want to stop smoking and intend to in the next 3 months*; (7) *I REALLY want to stop smoking and intend to in the next month.* A binary variable was derived to indicate high motivation to quit (yes (score of 6 or 7) vs no (score of 1–5)).

Having made a past-year quit attempt was also included as an additional adjustment in the quit attempt analysis as this has also been shown to be predictive of prospective quit attempts.[41] Participants were asked at baseline: *How many serious attempts to stop smoking have you made in the last 12 months? By serious attempt I mean you decided that you would try to make sure you never smoked again. Please include any attempt that you are currently making, and please include any successful attempt within the last 12 months.* Quit attempts were coded 1 for those who reported at least one quit attempt and 0 for those who reported no quit attempts in the last 12 months.

### Statistical analyses

The analysis plan was preregistered on the Open Science Framework (https://osf.io/w6cxd/) and all analyses conducted in IBM SPSS Statistics V.24.

To assess the representativeness of the analytic sample, we compared responders and non-responders to the 6-month follow-up survey on baseline smoking and sociodemographic characteristics using independent t-tests for continuous variables, and $\chi^2$ tests for categorical variables. Within the analysed sample, we also compared dual

e-cigarette users with dual OTC NRT users on the same characteristics.

We used linear regression to analyse the association between dual e-cigarette use versus dual OTC NRT use at baseline and (1) change in cigarettes per day, and log-binomial regression to analyse associations between e-cigarette use at baseline and (2) quit attempts and (3) use of cessation aids; with and without adjustment for the above-mentioned covariates. We also added an unplanned analysis following peer review which tested the association with use of behavioural support and/or prescription medication in the most recent quit attempt, in order to disentangle continuation of use of e-cigarettes or OTC NRT from a behavioural outcome that could be conceptualised as a more distinctive step toward cessation. In all models, dual OTC NRT use at baseline was the reference category.

As an unplanned sensitivity analysis, we replicated the adjusted quit attempt analysis substituting the full 7-level motivation to quit variable for the binary high/low motivation variable used in the primary analyses, to evaluate the robustness of this variable as an adjustment.

## RESULTS

Of 9798 current smokers surveyed at baseline between March 2014 and September 2016, 2318 (23.7%) reported dual e-cigarette use or dual OTC NRT use. Of the 2318 smokers eligible at baseline, 413 (17.8%) completed the 6-month follow-up survey. Responders were significantly older and more socioeconomically advantaged than non-responders, and they reported a higher average cigarettes per day (online supplementary table S1). There were no significant differences by sex, level of addiction, non-daily smoking, motivation to quit, past-year quit attempts or product used (e-cigarettes vs OTC NRT; online supplementary table S1).

Baseline characteristics of the analytic sample are shown in table 1. Dual e-cigarette users were on average significantly younger than dual NRT users, were slightly heavier smokers and were less likely to report high motivation to quit. The groups did not differ significantly on any other variable.

### Smoking reduction
Between baseline and 6-month follow-up, mean cigarette consumption reduced by 0.06 (SD 6.57) cigarettes per day in dual e-cigarette users and increased by 2.09 (5.88) cigarettes per day in dual OTC NRT users. This difference remained significant after adjustment for age, sex, social grade and level of nicotine addiction at baseline (table 2).

### Quit attempts
Between baseline and 6-month follow-up, 45.7% (n=186) made at least one serious attempt to quit smoking. The rate of quit attempts was significantly lower among dual e-cigarette users (41.8%, n=123) than dual OTC NRT

**Table 1** Baseline sample characteristics

| | Dual users of e-cigarettes (n=298) | Dual users of OTC NRT (n=115) | P value |
|---|---|---|---|
| **Sociodemographic characteristics** | | | |
| Age (years), % (n) | | | |
| 16–31 | 20.1 (60) | 10.4 (12) | 0.006 |
| 32–47 | 29.9 (89) | 20.9 (24) | – |
| 48–63 | 35.9 (107) | 47.0 (54) | – |
| ≥64 | 14.1 (42) | 21.7 (25) | – |
| Female, % (n) | 46.0 (137) | 51.2 (59) | 0.331 |
| Social grade C2DE, % (n) | 52.7 (157) | 58.3 (67) | 0.308 |
| **Smoking characteristics** | | | |
| Cigarettes per day, mean (SD) | 13.52 (9.56) | 11.82 (6.94) | 0.049 |
| Strength of urges to smoke (range 0–5), mean (SD) | 2.26 (1.03) | 2.13 (1.05) | 0.266 |
| Non-daily smoker, % (n) | 9.7 (29) | 10.4 (12) | 0.830 |
| High motivation to quit, % (n) | 23.2 (69) | 35.8 (41) | 0.010 |
| Attempted to quit in past year, % (n) | 48.6 (144) | 53.6 (60) | 0.375 |

Age was categorised by SD bands (16 years), with ≥80 collapsed into the 64–79 group due to low numbers.
OTC NRT, over-the-counter nicotine replacement therapy.

users (55.8%, n=63) even after adjustment for age, sex, social grade, motivation to stop smoking and past-year quit attempts at baseline (table 2). A sensitivity analysis treating motivation to quit as a 7-level variable produced similar results, but the difference was no longer statistically significant (risk ratio 0.82, 95% CI 0.66; 1.02, p=0.079).

### Use of cessation aids
Of those who made at least one serious attempt to quit over the follow-up period (n=186), the majority (80.6%, n=150) reported having used an evidence-based cessation aid during their most recent quit attempt at 6-month follow-up. Overall, use of cessation aids did not differ significantly between those using e-cigarette (82.9%, n=102) and those using OTC NRT at baseline (76.2%, n=48; table 2), although there were some differences in the type of cessation aids used.

Smokers who reported dual e-cigarette use at baseline were most likely to subsequently use e-cigarettes as a cessation aid (74.7%), followed by prescription medication (11.7%; prescription NRT 5.2%, bupropion 0.6%, varenicline 5.8%), face-to-face behavioural support (5.8%) and OTC NRT (5.8%). By contrast, smokers who reported dual OTC NRT use at baseline were most likely to subsequently use prescription medication as a cessation

**Table 2** Associations between use of e-cigarettes compared with use of over-the-counter nicotine replacement therapy and (1) smoking reduction, (2) quit attempts, (3) use of cessation aids and (4) use of behavioural support and/or prescription medication specifically over 6-month follow-up

| | Unadjusted | | | Adjusted* | | |
|---|---|---|---|---|---|---|
| | B (SE) | 95% CI | P value | B (SE) | 95% CI | P value |
| Change in cigarettes per day | −2.15 (0.81) | −3.74; −0.56 | 0.008 | −2.01 (0.82) | −3.62; −0.39 | 0.015 |
| | RR | 95% CI | P value | RR | 95% CI | P value |
| Quit attempts | 0.75 | 0.61; 0.93 | 0.008 | 0.82 | 0.67; 1.00 | 0.049 |
| Use of evidence-based cessation aids† | 1.04 | 0.92; 1.19 | 0.524 | 1.06 | 0.93; 1.21 | 0.388 |
| Use of behavioural support and/or prescription medication‡ | 0.26 | 0.15; 0.45 | <0.001 | 0.27 | 0.16; 0.47 | <0.001 |

*All models adjusted for age, sex and social grade. The model for change in cigarettes per day also adjusted for strength of urges to smoke; the model for quit attempts also adjusted for motivation to stop smoking and past-year quit attempts.
†Use of e-cigarettes, over-the-counter nicotine replacement therapy, behavioural support and/or prescription medication in the most recent quit attempt reported at follow-up.
‡Use of behavioural support and/or prescription medication in the most recent quit attempt reported at follow-up.
RR, risk ratio.

aid (39.5%; prescription NRT 30.3%, bupropion 2.6%, varenicline 6.6%), followed by OTC NRT (30.3%), face-to-face behavioural support (25.0%) and e-cigarettes (22.4%). When analysed separately from e-cigarettes and OTC NRT, dual e-cigarette users who made a serious quit attempt were significantly less likely than dual OTC NRT users who made a quit attempt to report using behavioural support or prescription medication during their most recent quit attempt at 6-month follow-up (table 2).

## DISCUSSION

In this study, we assessed prospective associations of dual use of e-cigarettes and cigarettes with smoking reduction, quit attempts, and use of evidence-based cessation aids compared with dual use of an alternative licensed nicotine product, OTC NRT. Dual e-cigarette users smoked on average two fewer cigarettes per day at 6-month follow-up than at baseline compared with dual OTC NRT users. However, dual e-cigarette use was associated with 18% reduced odds of making a serious quit attempt at 6-month follow-up relative to dual OTC NRT use. Although there was no significant difference in the use of evidence-based cessation aids by those who attempted to quit, dual e-cigarette use was associated with 73% reduced odds of using behavioural support or prescription medication specifically.

The association we observed between e-cigarette use and smoking reduction is in line with previous research that has documented a greater reduction in cigarette consumption associated with use of e-cigarettes compared with use of OTC NRT by smokers intending to quit,[32] and with use of e-cigarettes compared with non-use among smokers not intending to quit.[24] In our sample, dual e-cigarette users reported smoking more cigarettes per day than dual OTC NRT users at baseline. Even after adjustment for confounding variables, and exclusion of those who were non-smokers at follow-up (potentially causing

the effect size to be underestimated), dual e-cigarette users' mean cigarettes per day was significantly lower at 6-month follow-up than at baseline compared with dual OTC NRT users', leaving the two groups with similar consumption at follow-up. This finding does not support concerns that e-cigarettes perpetuate smoking, but rather may be associated with reduced smoking among heavier smokers.

While benefits of dual e-cigarette use on cigarette consumption lend support to e-cigarettes as a harm reduction strategy, it is possible that smokers who may have otherwise initiated a quit attempt settled for smoking reduction instead. Consistent with this, our results show that dual e-cigarette users had 18% lower odds than dual OTC NRT users to make a serious quit attempt over the 6-month follow-up period. This finding is in agreement with evidence that smokers using e-cigarettes without the intention to quit are less likely to attempt to quit,[42] but contrasts with previous real-world research that has found e-cigarettes to be more effective than OTC NRT in promoting cessation when used to support a quit attempt.[31] Importantly, the comparison in the present study was between dual e-cigarette use and dual OTC NRT use, rather than dual e-cigarette use and exclusive smoking. In a recent study that compared dual e-cigarette use, dual NRT use and exclusive smoking, a hierarchical association with quit attempts was observed: dual e-cigarette use was associated with a higher rate of quit attempts than exclusive smoking but a lower rate than NRT.[43] Taken together, the evidence appears to support concerns that e-cigarettes may divert cessation attempts,[44] but only relative to NRT.

A possible explanation is that, compared with NRT, e-cigarettes may reduce motivation to quit, for example, by allowing temporary abstinence in situations where smoking is prohibited.[45] Previous research has indicated that while many smokers report that e-cigarettes assist

in resisting the urge to smoke in such situations,[46] OTC NRT is less effective.[47 48] Alternatively, the difference in the rate of quit attempts may simply reflect the fact that dual e-cigarette users were less likely than dual OTC NRT users to be motivated to quit to begin with. In support of the importance of motivation in the association between dual e-cigarette use and quit attempts, while our primary analyses showed a significantly lower rate of quit attempts among e-cigarette users even after adjustment for a dichotomous (high/low) indicator of motivation to quit, this result was not significant in a sensitivity analysis using a more nuanced 7-level measure of motivation. Further research into the mediators of the relationship between e-cigarette use and quit attempts is required to shed light on the reasons for this potential adverse consequence of e-cigarette use and inform targeted interventions to prevent dual e-cigarette use from deterring smokers from quitting.

Despite dual e-cigarette users being significantly less likely than dual OTC NRT users to make a quit attempt, those who attempted to quit did not differ significantly in their use of evidence-based cessation aids. This suggests that dual e-cigarette use does not undermine their use of evidence-based support. With use of cessation aids a strong predictor of quit success,[14] this is important for maximising the chance that a quit attempt is successful. The finding that the majority (~80%) of those dual using e-cigarettes or OTC NRT subsequently made a quit attempt using evidence-based cessation aids is consistent with evidence that previous use of a cessation aid increases preference and likelihood of their future use.[35 36] To maximise statistical power in a limited sample, evidence-based cessation aids (e-cigarettes, NRT, face-to-face behavioural support and prescription medication) were combined for analysis, but descriptive data indicated that dual e-cigarette users tended to subsequently opt for e-cigarettes as a cessation aid, while dual OTC NRT users tended to use prescription medication or OTC NRT. Consistent with the fact that stop smoking services in England typically provide a combination of prescription medication and behavioural support, use of face-to-face behavioural support was substantially higher in the group who reported dual OTC NRT use at baseline. An unplanned analysis that focused on use of behavioural support or prescription medication revealed that dual e-cigarette users were substantially less likely than dual OTC NRT users to report use of these cessation aids. It is not clear from our results how far this reflects greater (perceived) effectiveness of e-cigarettes than OTC NRT or lower motivation to quit causing dual e-cigarette users being less likely to seek out alternative support. This is something that should be explored in future research.

A key strength of the study is the prospective design. However, there were also several limitations. Smoking reduction and quitting behaviour were examined only over a 6-month period, and thus replication of results with longer-term studies is required. The follow-up rate was also low, with just 17.8% of eligible baseline participants completing the 6-month follow-up survey, and significant differences were noted between responders and non-responders on several variables. Thus, the results may not be representative of all dual e-cigarette users in England and may not generalise to other samples. The reliance on self-reported data may increase measurement error or bias in key variables, such as quit attempts at 6-month follow-up. For example, research has found that recall of failed quit attempts is low,[49] which may reduce the observed effectiveness of harm reduction aids for quit attempts. If recall bias differs between dual users of e-cigarettes and OTC NRT, this may create spurious associations. Further, associations with use of support during quit attempts may be biased if unaided quit attempts were recalled less frequently than aided.[50] Additionally, while analyses were adjusted for several potential confounders, there is the possibility of residual confounding by unmeasured variables, such as the frequency[51] and duration[52] of e-cigarette/NRT use and the type of e-cigarette/NRT used. Future research with more detailed assessment of e-cigarette use could provide useful insights. Finally, this study was conducted in England, where e-cigarettes are regulated under the European Union's Tobacco Products Directive, which includes minimum standards for the safety and quality of all e-cigarettes and refill containers and restricts e-liquids to a nicotine strength of no more than 20 mg/mL. Our results may not generalise to other countries that do not have regulations for quality control or restrictions on nicotine concentration; particularly if higher levels of nicotine in e-cigarettes influence their effectiveness as a cessation aid.

## CONCLUSIONS

These findings lend some support to concerns that e-cigarettes may discourage quitting activity relative to other popular nicotine products, showing slightly lower odds of quit attempts among dual users of e-cigarettes and cigarettes compared with dual users of OTC NRT and cigarettes. However, they provide no evidence of an adverse effect of e-cigarette use on cigarette consumption; on the contrary, dual e-cigarette use was associated with greater smoking reduction compared with dual OTC NRT use. In addition, dual use of e-cigarettes did not appear to undermine use of evidence-based cessation aids during quit attempts but was associated with reduced odds of using behavioural support or prescription medication specifically.

**Contributors** EF, JB and LS conceived and designed the study. SJ and EF analysed the data and drafted the manuscript. JB and LS provided critical revisions. All authors read and approved the submitted version.

**Funding** Cancer Research UK funded data collection (C1417/A22962; C44576/A19501) and SJ and JB's salary (C1417/A22962). SJ's salary was also supported by the Economic and Social Research Council (ES/R005990/1). The funders had no final role in the study design; in the collection, analysis and interpretation of data; in the writing of the report or in the decision to submit the paper for publication.

All researchers listed as authors are independent from the funders and all final decisions about the research were taken by the investigators and were unrestricted.

**Competing interests** LS has received a research grant and honoraria for a talk and travel expenses from manufacturers of smoking cessation medications (Pfizer and Johnson & Johnson). JB has received unrestricted research funding from Pfizer, who manufacture smoking cessation medications. All authors declare no financial links with tobacco companies or e-cigarette manufacturers or their representatives.

**Patient and public involvement** Patients and/or the public were not involved in the design, or conduct, or reporting, or dissemination plans of this research.

**Patient consent for publication** Not required.

**Ethics approval** The University College London Ethics Committee provided ethical approval for the Smoking Toolkit Study (ID 0498/001).

**Provenance and peer review** Not commissioned; externally peer reviewed.

**Data availability statement** Data are available from the corresponding author on reasonable request.

**ORCID iD**
Sarah E Jackson http://orcid.org/0000-0001-5658-6168

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
