## [Reviewer comments · BMJ Open]

ARTICLE DETAILS

TITLE (PROVISIONAL)	Is dual use of nicotine products and cigarettes associated with smoking reduction and cessation behaviours? A prospective study in England
AUTHORS	Jackson, Sarah; Farrow, Emma; Brown, Jamie; Shahab, Lion

VERSION 1 – REVIEW

REVIEWER	Laurie Zawertailo Centre for Addiction and Mental Health, Canada
REVIEW RETURNED	16-Dec-2019

GENERAL COMMENTS	This paper describes a secondary analysis of survey data collected from the Smoking Toolkit Study in England. The aim of the paper was to examine the associations between use of e-cigarettes versus OTC NRT in current smokers at baseline and indicators of changes in smoking behaviour at 6-month follow-up including quit attempts and use of 'evidence-based cessation aids'. The rationale for this study as it is presented in the Introduction is not clear to me. If they are already using e-cigs or NRT at baseline (both of which are considered in this analysis to be evidence-based cessation aids') then does that not in itself indicate that people are using these aids to reduce their smoking in an effort to eventually quit - at least a significant proportion? If so, then how can you state that use of these aids at 6-months is indicative of anything different? The logic to me seems circular. This coupled with the fact that the response rate to the 6 month follow-up was extremely low at less than 18% does not give me a great deal of confidence that the findings are robust. As such, I am sorry that I must recommend rejection of this manuscript.
--

REVIEWER	Dale Mantey UTSPH, United States
REVIEW RETURNED	24-Dec-2019

GENERAL COMMENTS	Title: Prospective associations of dual use of nicotine products and cigarettes with smoking reduction and cessation behaviours in England General Comments: Overall, this is an exceptionally well-written and thoroughly researched manuscript. I've provided some recommendations for expanding the scope/research questions of this manuscript but those are purely recommendations. The authors may consider them or decline to incorporate them.
--

Beyond the research questions, there are serious concerns with the study sample. Descriptive statistics provided in Table 1 show systematic differences between e-cigarette users and OTC-NRT users. While this common for “real world” research such as prospective cohort studies, the differences in Table 1 are indicative of possible confounding as the e-cigarette users are younger (a predictor of study outcomes) and smoke more cigarettes per day (study outcome 1) while being less likely to have cessation intentions (study outcome 2). These systematic differences on such critical variables should, at the very least, inform the interpretation of study findings – which they currently do not. Specific comments for this are available below.

This is a good paper that could provide essential and direct implications for public health practice, research, and regulatory policy. Good job authors.

Introduction:

1. In the introduction (paragraph 3; Line 42), the authors should consider speaking more to the role of intentions and motivations as it pertains to cessation. In a general sense, intentions/motivation to quit is a study outcome so the authors should speak to how intentions/motivations are the greatest predictor of smoking cessation attempts (thus justifying exploring this as a study outcome). Specific to e-cigarettes, the authors should speak more to how many e-cigarette users may not be using for cessation and thus cessation intentions are important to understand. Further, two recent studies have shown that reasons for e-cigarette use are linked to differing impacts of e-cigarettes on cessation outcomes (listed below). The authors need not add a full literature review on motivations/intentions but at least provide some context to the importance of this outcome and the complexities of e-cigarettes.
 - a. Romijnders, K. A., Van Osch, L., De Vries, H., & Talhout, R. (2018). Perceptions and reasons regarding e-cigarette use among users and non-users: a narrative literature review. *International journal of environmental research and public health*, 15(6), 1190.
 - b. Mantey, D. S., Cooper, M. R., Loukas, A., & Perry, C. L. (2017). E-cigarette use and cigarette smoking cessation among Texas college students. *American journal of health behavior*, 41(6), 750-759.

Study Aims:

2. The third outcome, while interesting, could be vastly improved to be more informative of the differences between e-cigarettes and OTC-NRT. Currently, the third research question amounts to “which group is more likely to continue using a cessation method?” As currently coded, the third outcome could reflect continuing to use the same cessation type, switching cessation type (eg,

OTC-NTR user turning to e-cigarettes), or adding a new cessation type (eg, prescription medications). I recommend that the authors adjust this outcome to only encompass behavioural support or prescription medications, as this behavioral outcome could be conceptualized as a more intensive step towards cessation. It would also remove the confounding element outlined.

3. Similarly (but much less substantial) is that outcome one could be divided. The number of cigarettes is great as an outcome and should remain. However, the authors could also compare changes in “daily” smoking across the two groups. While this is a similar outcome to that of number of cigarettes smoked per day, it would be a meaningful contribution. The authors would then be exploring not only quantity (ie, cigarettes per day) but also frequency (ie, daily versus non-daily).

Methods

4. Very good job on the methods. A lot of moving parts here (eg, samples, outcomes, covariates, analyses, attrition) but the authors do an excellent job laying it all out for the reader to easily understand.

Results: The results reveal some areas of concern and possible bias/confounding. Specifically, the e-cigarette users seem to younger but smoke more cigarettes and have less intentions to quit at baseline than the OTC-NRT users. This suggests that there are systematic differences between e-cigarette users and OTC-NTR users in terms of age and smoking characteristics. This presents issues with statistical validity of the models as well as the interpretations of their findings.

5. First, number of cigarettes smoked per day is already higher among e-cigarette users (13.5) than OTC-NRT users (11.8) at baseline. So, effectively, using an e-cigarette brings the user down to OTC-NRT levels of cigarette smoking quantity? (ie, reducing their average of 13.5 at baseline to 11.5 at follow-up, which is equivalent to OTC-NRT users cigarettes per day at baseline of 11.8). That context should be added to the discussion to provide context to the findings.
 - a. Similarly, younger smokers traditionally smoke fewer cigarettes than older smokers. So, one would expect that the younger sample (e-cigarette users) would smoke fewer cigarettes than the older sample (OTC-NTR users) at baseline but that isn't the case for this sample. Why?

Discussion: The authors do a good job of interpreting study findings – rather than simply restating the results. However, the issues raised regarding systematic differences between the two groups within the study sample make the interpretation a bit incomplete. At baseline, the e-cigarette group seems more addicted to cigarettes (measured by quantity of use) and are even

	less likely to make a cessation attempt at 6-month follow-up (which could, conceivably, also reflect a propensity for dependence and long-term use). This should be added to the discussion. 6. A small note on the interpretation: the authors should speak to how these findings may not be representative to other regions that have differing regulations on e-cigarettes (particularly nicotine concentration). For example, there are currently no restrictions on nicotine levels for e-cigarettes in the United States and no regulations on quality control of the devices. Thus, findings of this study among a sample in a region that caps nicotine levels in e-cigarettes, may differ from other regions that don't have such regulations. Specifically, high levels of nicotine in e-cigarette may improve or even inhibit the effectiveness of these devices as cessation aids and thus findings from this study cannot be viewed as representative to other regions.
--	--

VERSION 1 – AUTHOR RESPONSE

Reviewer: 1

Laurie Zawertailo
Centre for Addiction and Mental Health, Canada

This paper describes a secondary analysis of survey data collected from the Smoking Toolkit Study in England. The aim of the paper was to examine the associations between use of e-cigarettes versus OTC NRT in current smokers at baseline and indicators of changes in smoking behaviour at 6-month follow-up including quit attempts and use of 'evidence-based cessation aids'.

The rationale for this study as it is presented in the Introduction is not clear to me. If they are already using e-cigs or NRT at baseline (both of which are considered in this analysis to be evidence-based cessation aids) then does that not in itself indicate that people are using these aids to reduce their smoking in an effort to eventually quit - at least a significant proportion? If so, then how can you state that use of these aids at 6-months is indicative of anything different? The logic to me seems circular. This coupled with the fact that the response rate to the 6 month follow-up was extremely low at less than 18% does not give me a great deal of confidence that the findings are robust. As such, I am sorry that I must recommend rejection of this manuscript.

Response: It has been argued that dual e-cigarette use may prevent cessation activities, i.e. maintain smoking (e.g. see <https://www.ncbi.nlm.nih.gov/pubmed/26776875>). The fact that dual use may well indicate an intention to reduce harm does not mean that this translates into action. It is therefore an empirical question whether dual e-cigarette use is associated with any future benefit, which is investigated here, contrasted with a positive control (dual NRT use).

In response to your comments and the suggestion of Reviewer 2, we have added an additional analysis exploring use of behavioural support or prescription medication specifically, independent of use of the aids smokers were already using at baseline.

We appreciate your concerns about the response rate. We do acknowledge this as an important limitation of the study with appropriate caveats about potential issues with representativeness and generalisability. However, given the scarcity of prospective evidence in this area we do believe these results make a useful contribution to the literature and can be interpreted in light of the study's limitations.

Reviewer: 2

Dale Mantey

Title: Prospective associations of dual use of nicotine products and cigarettes with smoking reduction and cessation behaviours in England

General Comments: Overall, this is an exceptionally well-written and thoroughly researched manuscript. I've provided some recommendations for expanding the scope/research questions of this manuscript but those are purely recommendations. The authors may consider them or decline to incorporate them.

Beyond the research questions, there are serious concerns with the study sample. Descriptive statistics provided in Table 1 show systematic differences between e-cigarette users and OTC-NRT users. While this common for "real world" research such as prospective cohort studies, the differences in Table 1 are indicative of possible confounding as the e-cigarette users are younger (a predictor of study outcomes) and smoke more cigarettes per day (study outcome 1) while being less likely to have cessation intentions (study outcome 2). These systematic differences on such critical variables should, at the very least, inform the interpretation of study findings – which they currently do not. Specific comments for this are available below.

This is a good paper that could provide essential and direct implications for public health practice, research, and regulatory policy. Good job authors.

Response: Thank you for your considered review. We outline the changes we have made in detail below.

Introduction:

1. In the introduction (paragraph 3; Line 42), the authors should consider speaking more to the role of intentions and motivations as it pertains to cessation. In a general sense, intentions/motivation to quit is a study outcome so the authors should speak to how intentions/motivations are the greatest predictor of smoking cessation attempts (thus justifying exploring this as a study outcome). Specific to e-cigarettes, the authors should speak more to how many e-cigarette users may not be using for cessation and thus cessation intentions are important to understand. Further, two recent studies have shown that reasons for e-cigarette use are linked to differing impacts of e-cigarettes on cessation outcomes (listed below). The authors need not add a full literature review on motivations/intentions but at least provide some context to the importance of this outcome and the complexities of e-cigarettes.

Response: While we do control for motivation to stop smoking in our analyses, this is not a specific outcome of our analyses (which focus on smoking reduction, quit attempts, and use of cessation support). As such, we have opted not to detract too much from the focus of our introduction with additional content on motivation, but have drawn attention to the suggested literature where we discuss evidence of changes in cigarette consumption among e-cigarette users not intending to quit:

"Experimental studies have shown similar effects even in smokers not intending to quit 23,24. This is notable given that many dual e-cigarette users do not use e-cigarettes for cessation 25,26."

We also return to the issue of motivation in the discussion, which we have now expanded: "A possible explanation is that, compared with NRT, e-cigarettes may reduce motivation to quit, for example, by allowing temporary abstinence in situations where smoking is prohibited⁴⁵. Previous research has indicated that while many smokers report that e-cigarettes assist in resisting the urge to smoke in such situations⁴⁶, OTC NRT is less effective^{47,48}. Alternatively, the difference in the rate of quit attempts may simply reflect the fact that dual e-cigarette users were less likely than dual OTC NRT users to be motivated to quit to begin with. In support of the importance of motivation in the association between dual e-cigarette use and quit attempts, while our primary analyses showed a significantly lower rate of quit attempts among e-cigarette users even after adjustment for a dichotomous (high/low) indicator of motivation to quit, this result was not significant in a sensitivity analysis using a more nuanced 7-level measure of motivation."

- a. Romijnders, K. A., Van Osch, L., De Vries, H., & Talhout, R. (2018). Perceptions and reasons regarding e-cigarette use among users and non-users: a narrative literature review. *International journal of environmental research and public health*, 15(6), 1190.
- b. Mantey, D. S., Cooper, M. R., Loukas, A., & Perry, C. L. (2017). E-cigarette use and cigarette smoking cessation among Texas college students. *American journal of health behavior*, 41(6), 750-759.

Study Aims:

2. The third outcome, while interesting, could be vastly improved to be more informative of the differences between e-cigarettes and OTC-NRT. Currently, the third research question amounts to "which group is more likely to continue using a cessation method?" As currently coded, the third outcome could reflect continuing to use the same cessation type, switching cessation type (eg, OTC-NTR user turning to e-cigarettes), or adding a new cessation type (eg, prescription medications). I recommend that the authors adjust this outcome to only encompass behavioural support or prescription medications, as this behavioral outcome could be conceptualized as a more intensive step towards cessation. It would also remove the confounding element I outlined.

Response: We appreciate this suggestion and now present this additional analysis in addition to our planned analyses:

Method: "We also added an unplanned analysis following peer review which tested the association with use of behavioural support and/or prescription medication in the most recent quit attempt, in order to disentangle continuation of use of e-cigarettes or OTC NRT from a behavioural outcome that could be conceptualised as a distinctive step toward cessation."

Results: "When analysed separately from e-cigarettes and OTC NRT, dual e-cigarette users who made a serious quit attempt were significantly less likely than dual OTC NRT users who made a quit attempt to report using behavioural support or prescription medication during their most recent quit attempt at 6-month follow-up (Table 2)."

Discussion: "An unplanned analysis that focused on use of behavioural support or prescription medication revealed that dual e-cigarette users were substantially less likely than dual OTC NRT users to report use of these cessation aids. It is not clear from our results how far this reflects greater (perceived) effectiveness of e-cigarettes than OTC NRT or lower motivation to quit causing dual e-cigarette users being less likely to seek out alternative support. This is something that should be explored in future research."

Conclusion: "In addition, dual use of e-cigarettes did not appear to undermine use of evidence-based cessation aids during quit attempts but was associated with reduced odds of using behavioural support or prescription medication specifically."

3. Similarly (but much less substantial) is that outcome one could be divided. The number of cigarettes is great as an outcome and should remain. However, the authors could also compare changes in "daily" smoking across the two groups. While this is a similar outcome to that of number of cigarettes smoked per day, it would be a meaningful contribution. The authors would then be exploring not only quantity (ie, cigarettes per day) but also frequency (ie, daily versus non-daily).

Response: We have considered this suggestion but believe analysing change in daily smoking over a 6-month period would be quite difficult to interpret. If the whole sample is analysed, one might test the odds of daily smoking at follow-up adjusting for whether they smoked daily at baseline, but the coefficients would conflate changes from daily to non-daily and non-daily to daily. Alternatively, one would need to perform separate analyses on those who were daily and non-daily smokers at baseline and explore the odds of change in status, but this would restrict the already quite small numbers of participants in each group resulting in a loss of statistical power. As such, we prefer not to add this additional analysis.

Methods

4. Very good job on the methods. A lot of moving parts here (eg, samples, outcomes, covariates, analyses, attrition) but the authors do an excellent job laying it all out for the reader to easily understand.

Response: Thank you.

Results: The results reveal some areas of concern and possible bias/confounding. Specifically, the e-cigarette users seem to younger but smoke more cigarettes and have less intentions to quit at

baseline than the OTC-NRT users. This suggests that there are systematic differences between e-cigarette users and OTC-NTR users in terms of age and smoking characteristics. This presents issues with statistical validity of the models as well as the interpretations of their findings.

Response: It is for this reason that we include adjusted regression coefficients in addition to our unadjusted models, and focus our interpretation and discussion on the adjusted results. See responses below for specific amendments we have made in line with your comments on confounding.

5. First, number of cigarettes smoked per day is already higher among e-cigarette users (13.5) than OTC-NRT users (11.8) at baseline. So, effectively, using an e-cigarette brings the user down to OTC-NRT levels of cigarette smoking quantity? (ie, reducing their average of 13.5 at baseline to 11.5 at follow-up, which is equivalent to OTC-NRT users cigarettes per day at baseline of 11.8). That context should be added to the discussion to provide context to the findings.

Response: We have edited the wording of our discussion of these results to add this contextual information:

“In our sample, dual e-cigarette users reported smoking more cigarettes per day than dual OTC NRT users at baseline. Even after adjustment for confounding variables, and exclusion of those who were non-smokers at follow-up (potentially causing the effect size to be underestimated), dual e-cigarette users’ mean cigarettes per day was significantly lower at 6-month follow-up than at baseline compared with dual OTC NRT users’, leaving the two groups with similar consumption at follow-up. This finding does not support concerns that e-cigarettes perpetuate smoking, but rather may be associated with reduced smoking among heavier smokers.”

- a. Similarly, younger smokers traditionally smoke fewer cigarettes than older smokers. So, one would expect that the younger sample (e-cigarette users) would smoke fewer cigarettes than the older sample (OTC-NTR users) at baseline but that isn’t the case for this sample. Why?

Response: While e-cigarette users tend to be younger than NRT users, there are other variables implicated in the association between e-cigarette use and consumption, including that e-cigarette users tend to be significantly more addicted to nicotine, and those who are more addicted typically smoke more cigarettes per day.

Discussion: The authors do a good job of interpreting study findings – rather than simply restating the results. However, the issues raised regarding systematic differences between the two groups within the study sample make the interpretation a bit incomplete. At baseline, the e-cigarette group seems more addicted to cigarettes (measured by quantity of use) and are even less likely to make a cessation attempt at 6-month follow-up (which could, conceivably, also reflect a propensity for dependence and long-term use). This should be added to the discussion.

Response: As mentioned in response to an earlier comment, we have added contextual information on baseline differences in cigarette consumption between groups:

“In our sample, dual e-cigarette users reported smoking more cigarettes per day than dual OTC NRT users at baseline. Even after adjustment for confounding variables, and exclusion of those who were non-smokers at follow-up (potentially causing the effect size to be underestimated), dual e-cigarette users’ mean cigarettes per day was significantly lower at 6-month follow-up than at baseline compared with dual OTC NRT users’, leaving the two groups with similar consumption at follow-up. This finding does not support concerns that e-cigarettes perpetuate smoking, but rather may be associated with reduced smoking among heavier smokers.”

We have also added to our discussion of the potential role of motivation in driving differences in quit attempts:

“A possible explanation is that, compared with NRT, e-cigarettes may reduce motivation to quit, for example, by allowing temporary abstinence in situations where smoking is prohibited⁴⁵. Previous research has indicated that while many smokers report that e-cigarettes assist in resisting the urge to smoke in such situations⁴⁶, OTC NRT is less effective^{47,48}. Alternatively, the difference in the rate of quit attempts may simply reflect the fact that dual e-cigarette users were less likely than dual OTC NRT users to be motivated to quit to begin with. In support of the importance of motivation in the association between dual e-cigarette use and quit attempts, while our primary analyses showed a significantly lower rate of quit attempts among e-cigarette users even after adjustment a dichotomous (high/low) indicator of motivation to quit, this result was not significant in a sensitivity analysis using a more nuanced 7-level measure of motivation.”

6. A small note on the interpretation: the authors should speak to how these findings may not be representative to other regions that have differing regulations on e-cigarettes (particularly nicotine concentration). For example, there are currently no restrictions on nicotine levels for ecigarettes in the United States and no regulations on quality control of the devices. Thus, findings of this study among a sample in a region that caps nicotine levels in e-cigarettes, may differ from other regions that don't have such regulations. Specifically, high levels of nicotine in e-cigarette may improve or even inhibit the effectiveness of these devices as cessation aids and thus findings from this study cannot be viewed as representative to other regions.

Response: We appreciate this suggestion and have added the following to the end of our limitations section:

“Finally, this study was conducted in England, where e-cigarettes are regulated under the European Union’s Tobacco Products Directive, which includes minimum standards for the safety and quality of all e-cigarettes and refill containers and restricts e-liquids to a nicotine strength of no more than 20mg/ml. Our results may not generalise to other countries that do not have regulations for quality control or restrictions on nicotine concentration; particularly if higher levels of nicotine in e-cigarettes influence their effectiveness as a cessation aid.”

VERSION 2 – REVIEW

REVIEWER	Dale Mantey UTSPH, United States
REVIEW RETURNED	03-Feb-2020
GENERAL COMMENTS	The authors do a good job of responding to all areas of concern. Very good contribution to the field.